# RETRACTED: Protective Effects of Astaxanthin Supplementation against Ultraviolet-Induced Photoaging in Hairless Mice

**DOI:** 10.3390/biomedicines8020018

**Published:** 2020-01-21

**Authors:** Xing Li, Tomohiro Matsumoto, Miho Takuwa, Mahmood Saeed Ebrahim Shaiku Ali, Takumi Hirabashi, Hiroyo Kondo, Hidemi Fujino

**Affiliations:** Department of Rehabilitation Science, Kobe University Graduate School of Health Sciences, 7-10-2 Tomogaoka, Suma-ku, Kobe 654-0142, Hyogo, Japan; lixing3633@gmail.com (X.L.); tomo08047777@gmail.com (T.M.); miho.takuwa@gmail.com (M.T.); mahmood_6195@hotmail.com (M.S.E.S.A.); hirabayashitkm1@gmail.com (T.H.); hiroyokondo@gmail.com (H.K.)

**Keywords:** astaxanthin, antioxidant, skin, ultraviolet, photoaging, capillary

## Abstract

Ultraviolet (UV) light induces skin photoaging, which is characterized by thickening, wrinkling, pigmentation, and dryness. Astaxanthin (AST), a ketocarotenoid isolated from *Haematococcus pluvialis*, has been extensively studied owing to its possible effects on skin health as well as UV protection. In addition, AST attenuates the increased generation of reactive oxygen species (ROS) and capillary regression of the skeletal muscle. In this study, we investigated whether AST could protect against UV-induced photoaging and reduce capillary regression in the skin of HR-1 hairless mice. UV light induces wrinkle formation, epidermal thickening, and capillary regression in the dermis of HR-1 hairless mice. The administration of AST reduced the UV-induced wrinkle formation and skin thickening, and increased collagen fibers in the skin. AST supplementation also inhibited the generation of ROS, decreased wrinkle formation, reduced epidermal thickening, and increased the density of capillaries in the skin. We also found an inverse correlation between wrinkle formation and the density of capillaries. An association between photoaging and capillary regression in the skin was also observed. These results suggest that AST can protect against photoaging caused by UV irradiation and the inhibitory effects of AST on photoaging may be associated with the reduction of capillary regression in the skin.

## 1. Introduction

The skin is the largest organ in mammals. It predominantly consists of an external stratified, non-vascularized epithelium (epidermis), the underlying connective tissue (dermis), and the subcutaneous adipose tissue known as the hypodermis. The skin can regulate local and global homeostasis by sensing the environment [1,2]. However, prolonged exposure to ultraviolet (UV) radiation causes photoaging, which is clinically characterized by dryness, deep wrinkling, laxity, and pigmentation [3,4].

A recent study showed that exposure of the skin to UV radiation generates reactive oxygen species (ROS), accompanied by the expression of genes and proteins leading to photo-damage and photo-carcinogenesis [5,6]. ROS also cause degradation of the dermal collagen and elastic fibers [7]. It has been established that oxidative stress initiated by ROS, as well as DNA damage, are induced in skin cells upon UV irradiation; this damage eventually leads to premature skin photoaging. Thus, UV irradiation results in fine wrinkles due to the diminished or defective synthesis of collagen and elastin in the dermis [4,8].

Astaxanthin (AST), which is naturally found in seafood, such as shrimp and salmon, as well as in *Haematococcus pluvialis* (*H. pluvialis*), is a red-colored pigment that belongs to the xanthophyll subclass of carotenoids and that can scavenge ROS [9]. AST possesses many highly potent pharmacological effects. These include protection against UV-induced cell damage and chronic inflammatory diseases, promotion of immunomodulatory activities, alleviation of metabolic syndromes, cardioprotective effects, antidiabetic activity, inhibition of neuronal damage, anti-aging effects on the skin, and anticancer activity as well as the suppression of cell membrane peroxidation [5,10]. AST has been shown to improve dermal health by direct and downstream effects at different steps of the oxidative stress cascad while simultaneously inhibiting inflammatory mediators [11]. In addition, several clinical studies have reported the effects of AST against photoaging in the skin [12]. AST supplementation has been shown to cause significant anti-aging and functional improvements in the skin of healthy women [13]. It has also been reported that AST supplementation improves the appearance of crow’s feet wrinkles, enhances elasticity, and reduces transepidermal water loss in the skin of healthy women [14]. A previous study reported that dietary AST administered at 100 mg/kg/day to hairless mice for 10 days caused significant antiaging results [15]. Thus, AST has profound value. However, the effect of AST on the capillaries in the skin has rarely been reported. Therefore, we aimed to investigate whether AST could prevent UV-induced epidermal thickening and ROS generation and reduce capillary regression in the skin of HR-1 hairless mice in this study.

## 2. Materials and Methods

### 2.1. Animals and Astaxanthin Source

The mice were housed individually under a 12/12 h light/dark cycle at room temperature (22 ± 2 °C) and 50% relative humidity. Wood chips were used as bedding in the cages and replaced every 3 days. HR-1 Hairless mice were obtained from the Japan SLC, Inc.; they were fed a commercial diet (CE-2, CLEA Japan, Inc.) and allowed access to tap water ad libitum throughout the study. AST oil was provided by Fuji Chemical Industry Co. Ltd., Toyama, Japan.

### 2.2. Astaxanthin Treatment and Experimental Design

The HR-1 hairless mice were randomly divided into three groups (*n* = 8 per group): (1) Control, olive oil alone (CON); (2) UV-induced plus olive oil (UV); and (3) UV-induced plus AST (AstaReal Oil 50F, Fuji Chemical Industry Co. Ltd., Toyama, Japan). In the CON group, the mice were orally administered the olive oil only (dosages of olive oil 100 mg/kg body weight) by gavage daily. In the UV and AST group, the mice were orally supplemented with the mixture of AST (dosage of AST 100 mg/kg body weight). This study was approved by the Institution Animal Care and Use Committee; the experimental protocol followed the Kobe University Animal Experimentation Regulations (Kobe, Japan, approval number: p180802, 8 August 2018). All experiment and animal care programs were managed in conformance with the Guide for the Care and Use of Laboratory Animals published by the US National Institutes of Health (NIH publication no. 85–23, revised 996).

### 2.3. UV Irradiation

UV irradiation of mice was performed using a UV lamp (Lutron UV-340A, Taipei, Taiwan) with a spectral region between 365 with 400 nm. A UV meter was used to measure the UV radiation. The distance from the UV lamp to the skin on the back of the mice was maintained at 20 cm. The mice were exposed to UV lamps for 8 weeks and the exposure intensity increased gradually from 1 to 4 minimal erythema dose (MED) (equivalent to 100 mJ/cm^2^). The skin on the back of the mice was irradiated with UV light three times per week. The mice were irradiated with UV at 1 MED intensity during the first to second week, with 2 MED intensity during the third to fourth week, with 3 MED intensity during the fifth to sixth week, and with 4 MED intensity during the seventh to eighth week [16]. The body weight of HR-1 hairless mice undergoing different treatments, and the food and water consumption were recorded every day.

### 2.4. Wrinkle Measurement

Skin condition was assessed by taking photographs of the mouse dorsal skin at the end of week 8 to confirm the extent of wrinkle formation. To estimate the proportion of wrinkle area to skin area, a square of 2 cm^2^ centered on the spine was taken. Replicas were prepared using a SILFLO kit (CuDerm Corporation, Dallas, TX, USA) to scale the area of the wrinkles just before harvesting the skin of the mice.

### 2.5. Hematoxylin Eosin (H&E) Staining for Epidermal Thickness

The dorsal skin of the mice located between the ilia was harvested at the end of the experiment, at 16 weeks after the application of anesthesia; the mice were then killed with an overdose of sodium pentobarbital. The skin was fixed with 4% paraformaldehyde (PFA) for 24 h, and 30-μm frozen sections were taken. Histological features and epidermal thickness were examined by H&E staining. The mounting medium (Vector Laboratories, Inc., Burlingame, CA, USA) was used for mounting the sections onto slides. The images were taken by imager microscope (Japan). Automatic measurement analysis of each staining was implemented by IMT i-solution (IMT i-solution Inc., Vancouver, BC, Canada).

### 2.6. Masson’s Trichrome Staining for Collagen Intensity

Collagen staining was performed at the end of the experiment, and the dorsal skin between the ilia of the mice was collected under anesthesia at the end of the experiment (16 weeks). The mice were killed with an overdose of sodium pentobarbital. The skin was dipped and immobilized in 4% PFA for 24 h, and 30-μm frozen sections were collected. To determine the density of collagen fibers, Masson’s trichrome staining was performed. Mounting medium was used for mounting the slides. The images were taken by imager microscope (Japan). Each staining was automatically measured and analyzed using ImageJ software (NIH, Bethesda, MD, USA).

### 2.7. Assessment of ROS

In situ generation of ROS was evaluated using oxidative fluorescent dihydroethidium (DHE), which emits light when it interacts with O_2_ to form oxyethidium. Dihydroethidium is a cell-permeable agent that interacts with nucleic acids to emit red light that is qualitatively detectable by fluorescence microscopy (BX51; Olympus, Tokyo, Japan) with a rhodamine filter (excitation, 490 nm; emission, 590 nm). This staining method has been previously used to determine ROS activity [17,18]. Briefly, the sections (30 μm thick) were incubated with 5 × 10^−6^ mol/L DHE (Wako Pure Chemicals, Osaka, Japan) for 30 min at 37 °C in a dark box, rinsed with 37 °C PBS, and observed using a fluorescence microscope (filter with excitation at 545 nm). Densitometric analysis of DHE fluorescence was performed with Image J using five images per section of skin, and the results were reported as a percentage of the control group. The densitometric analyses of fluorescence were obtained using blind analysis for each experimental group. There are limitations with this technique [19,20]. For example, freezing and cutting muscle sections that are warmed in the presence of DHE could artificially release reactive oxygen species. Though, this was minimized because all the tissues were prepared in a similar manner.

### 2.8. Alkaline Phosphatase Staining

The sections were stained with alkaline phosphatase (AP) to observe the capillarity of the skin [21]. The sections were incubated in 5-bromo-4-chloro-3-indolyl phosphate/nitro blue tetrazolium for 45 min at 37 °C and fixed with 4% paraformaldehyde. The number of capillaries was determined by counting the capillaries on each cryosection using microscopic images.

### 2.9. Statistical Analysis

The data are presented as the mean ± SD. The statistical analyses were carried out using one-way ANOVA, followed by the Scheffe’s post-hoc test. The SigmaStat statistical program (version11.2; Systat Software, San Jose, CA, USA) was used for statistical analysis of the data. Differences are considered significant at *p* < 0.05.

## 3. Results

### 3.1. Effects of AST on Wrinkle Formation in UV-Induced HR-1 Hairless Mice

In the present study, body weights were measured for determining the toxicity of the treatment. We found that there was no significant difference in the body weight between the CON group, UV group, and UV +AST group. At 8 weeks after UV exposure, skin conditions revealed by photography showed that the dorsal skin of UV-induced mice was lumpy and flaky compared with that of the control mice (Figure 1B). Nevertheless, flakiness and roughness were decreased in UV-induced mice treated with AST compared to that in mice exposed to UV irradiation alone. An analysis of wrinkle formation in silicon replicas revealed that the wrinkles of the control mice were thin and shallow (Figure 1A) while those of the UV-induced mice were thick and deep. Moreover, the UV-induced mice showed an increase in the percentage (approximately 1.6 times) area of wrinkles compared with the control mice (Figure 1C). These changes were reduced by AST treatment, which was reduced by 50% compared with that of the UV group. Analysis showed that AST reduces UV-induced wrinkle formation.

### 3.2. Effect of AST on Epidermal Thickness in the Dorsal Skin of UV-Induced Mice

In this study, we investigated UV-induced skin photoaging using the HR-1 hairless mouse model. To investigate the effects of AST on the photoaging of skin in vivo, the HR-1 mice were irradiated with UV. H&E staining revealed the effects of AST on histological changes in the dorsal skin (Figure 2A). The UV-induced mice had thicker epidermal layers than the mice that were not induced by UV, and the average thickness of the epidermis (CON group) increased from 80 to 120 μm (UV group). Nevertheless, the control mice and the AST-treated mice had thinner epidermal layers (approximately 45 μm) than the mice that were only irradiated with UV (Figure 2B).

### 3.3. Effect of AST on the Density of Collagen Fibers in the Dorsal Skin of UV-Irradiated HR-1 Hairless Mice

The changes in collagen in the dermis of HR-1Hairless mice were investigated in UV-irradiated mice treated with AST. Masson’s trichrome staining showed the effects of AST by histological changes in the dorsal skin (Figure 3A). We found a significant reduction in the density of collagen fibers in the dermis of UV-irradiated HR-1 hairless mice compared to that in the control. The density of collagen fibers dropped by approximately 20%, and treatment with AST decreased the UV irradiation-induced loss of collagen fibers (Figure 3B). The density of collagen fibers increased by approximately 20% compared with that of the UV group. These results show that AST can reduce UV irradiation-induced skin collagen fiber loss, indicating the protective effect of AST against UV-induced skin damage.

### 3.4. Effect of AST on Changes in the Density of Capillary Vessels in the Dorsal Skin of UV-Induced Mice

In order to investigate the changes of the density capillaries in the dorsal skin of UV-induced mice, AP staining was used to evaluate the presence and distribution of capillaries. As shown in Figure 4A, we also found that the results show that the density of capillary vessels is reduced in the dorsal skin of UV-induced mice. The density of the capillary vessels of the UV group decreased by approximately 52% compared with that of the CON group. Whereas the changes can be reversed by oral AST, the density of capillary vessels in the UV + AST group increased by approximately four times compared with that in the UV group. This result shows that AST attenuates UV-induced capillary regression.

### 3.5. Effect of AST on ROS Activity in the Dermis and Capillary Vessels in the Dorsal Skin of UV-Induced Mice

A marked enhancement in total ROS activity, as measured using DHE, was observed in UV-induced HR-1 hairless mice compared to that in control HR-1 hairless mice (Figure 5A). AST significantly decreased UV-induced ROS formation (Figure 5B). We determined the distribution of ROS in the skin and noted that UV-induced HR-1 hairless mice exhibited significantly increased ROS generation (the total amount of ROS is approximately three times compared with that of the CON group), not only in the epidermis but also in the dermis. The significant reduction in ROS in the skin, especially in the epidermis, blood vessels, cellular components of the dermis, and the extracellular matrix, shows that AST reduces UV-induced generation of ROS (the ROS level restores to CON level in the UV + AST group).

### 3.6. Correlation between the Density of Capillaries and Epidermal Thickness, Density of Collagen, and ROS Expression in the Dorsal Skin of UV-Induced Mice

We also investigated the relationship between capillaries, wrinkles, epidermal thickness, collagen, and ROS. We found that the density of capillaries is negatively correlated with the density of wrinkles, the thickness of epidermis, and the ROS levels (Figure 6A,C,D). The density of capillaries is positively correlated with the density of the collagen fibers (Figure 6B). Evidently, changes in the density of capillaries have an important effect on the photoaging of dorsal skin in UV-induced mice.

## 4. Discussion

The novel findings of this study are: (i) Capillary regression was observed in the UV-induced photoaging skin of HR-1 hairless mice, and this was accompanied by an increase in ROS generation; (ii) AST supplementation decreased skin photoaging, which is characterized by reduced skin thickening and the density of wrinkling, and maintained ROS levels and the density of capillaries at control levels; and (iii) changes in the density of capillaries were observed in the skin of HR-1 hairless mice, which has a close correlation with the density of wrinkles, density of collagen, epidermal thickness, and ROS levels. Thus, AST treatment was demonstrated as a protective therapy for skin photoaging.

Changes in the skin are the most prominent and visible signs of aging. Skin aging can be divided into intrinsic or chronologic aging, which is the process of senescence that affects all body organs, and extrinsic aging, e.g., photoaging, which occurs because of exposure to environmental factors [22]. Intrinsic (genetically determined) and extrinsic (UV- and toxic exposure-mediated) skin aging processes overlap and are strongly related to the increased generation of free radicals in the skin [23]. One of the most important factors influencing intrinsic aging is a gradual loss of function or degeneration that occurs at the cellular level [24].

Recently, the development of wrinkles in mice after UV exposure has been considered as an indicator of chronic UV exposure or photoaging [25]. Bissett et al. reported slight wrinkles after 5 weeks of UV exposure and permanent wrinkles after 15 weeks in HR-1 hairless mice [26]. Our study indicates that HR-1 hairless mice also display significant wrinkles after 8 weeks of UV exposure (Figure 1A).

UV leads to an increase in the epidermal thickness, termed hyperkeratosis, and induces damage response pathways in keratinocytes. After several hours of UV exposure, robust proliferation of epidermal keratinocytes occurs, which is mediated by a variety of epidermal growth factors [27]. In this study, we also found that dietary supplementation with AST for 8 weeks effectively prevented photoaging, including reduced wrinkle formation (Figure 1) and a decrease in the epidermal thickness in the dorsal skin of mice exposed to UV irradiation (Figure 2A,B); this effect may relate to the suppression of collagen breakdown by AST in the skin. Furthermore, Shin et al. noted an inverse correlation between wrinkling and important antioxidant enzymes that reduce the cellular levels of ROS in hairless mice [28].

UV may inactivate carotenoids in the skin and promote degradation of dermal collagen and elastin in a human skin model [29,30]. In this study, the HR-1 hairless mice were exposed to UV irradiation for 8 weeks and the collagen density decreased in the dorsal skin (Figure 3A). In the AST treatment group, however, the collagen density in the skin was not reduced and was higher compared to the control group (Figure 3B). UV radiation has been reported to promote the activation of enzymes that destroy elastic fibers and collagen, making the skin more prone to wrinkling in human facial skin [31]. A low concentration of ROS is needed to initiate the normal repair process of collagen. AST was shown to preserve this physiological function by redox regulation [32]. AST can effectively suppress in vitro cell damage caused by free radicals and the induction of (matrix metalloproteinase-1) MMP-1 protein in human dermal fibroblasts after UV irradiation [33]. AST has been reported to increase the expression of collagen by inhibiting the expression of MMP-1 and MMP-9 protein in an animal model [34]. In addition, AST increased the collagen content through inhibition of MMP-1 and MMP-3 protein expression in human dermal fibroblasts [35]. Hyun-SunYoon et al. have reported that dietary AST supplementation for 12 weeks increased collagen levels in human skin [36].

The exposure of the skin to UV irradiation induces the generation of reactive oxygen/nitrogen species or oxidative stress, which is capable of oxidizing lipids, proteins, and DNA. The resulting oxidized products, including lipid hydroperoxides, protein carbonyls, and 8-hydroxydeoxyguanosine, have been implicated in the onset of skin aging [37,38]. It has been reported that ROS are produced by various biochemical reactions in mitochondria, peroxisomes, and endoplasmic reticulum in cells or organelles [39,40]. Thus, UV induces ROS, primarily formed via oxidative cell metabolism, and plays a major role in both chronological aging and photoaging of skin [41]. This is consistent with our findings (Figure 5A). Camera et al. have reported that AST inhibits ROS formation and modulates the expression of oxidative stress-responsive enzymes, such as heme oxygenase-1 (HO-1), which is a marker of oxidative stress and a regulatory mechanism involved in cell adaptation against oxidative damage in the human dermal fibroblasts model [42]. In this study, we also found that dietary supplementation with AST for 8 weeks effectively prevented an ROS increase (Figure 5B).

In addition, it has been reported that AST decreases the level of oxidative stress, as indicated by reduced plasma malondialdehyde levels, and reverses age-related changes in the residual skin surface components of middle-aged subjects [43].

Skin capillaries serve an important function by supplying nutrients to the skin. A reduced density of peritubular capillaries was observed in mice subjected to unilateral ureteral obstruction; the density was significantly increased by AST treatment [44]. In addition, a previous study revealed that AST could prevent capillary regression in atrophied soleus muscles by upregulating vascular endothelial growth factor (VEGF) protein and downregulating thrombospondin-1 (TSP-1) protein in the soleus muscle of rats [45]. It has also been reported that reductions in the capillary diameter and volume are associated with vascular endothelial cell apoptosis via increased oxidative stress in hind limb unweighting of a rat model [46]. The data from the present study also indicate that the number of capillaries is markedly reduced after exposure to UV, whereas oral administration with AST prevented the decrease in capillaries in the skin of mice (Figure 4).

Furthermore, in UV-induced skin, the dermal thickness increased while the capillary density decreased, as recently reported on UV-induced photoaging in an in vitro and in vivo model by Deng et al. [20]. We also found an inverse correlation between wrinkling and the levels of ROS and the capillary number in the skin of mice (Figure 6). Thus, photoaging may be associated with capillary regression in the skin.

In the human brain microvascular endothelial cell line and rat aortic smooth muscle cell, it has been reported that capillary proliferation is related to ROS level regulation, angiogenesis, and ROS signals that regulate the formation of new blood vessels [47]. However, the relationship between the mechanism of capillary density degradation and AST was not explained in this experiment and would therefore benefit from further study.

Overall, our study indicates the protective effects of dietary AST against photoaging induced by UV radiation, such as reducing wrinkling, reduced thickness of the epidermis, and maintaining collagen density and the density of capillaries in the skin. We found that the density of capillaries is negatively correlated with the density of wrinkles, the thickness of the epidermis, and ROS levels. We also showed that the density of capillaries is positively correlated with the density of collagen fibers.

We found that dietary AST may prevent the effects of UV irradiation on photoaging, as well as on the dermal capillaries. However, the cumulative intake of AST in the skin and blood was not measured, nor did we ascertain the mechanism by which AST inhibits capillary regression. Petri and Lundebye reported that the accumulation of dietary AST in the hairless skin of the tail was much higher than those in other tissues [48]. Color changes on the tail skin by reflectance measurements using a portable spectrophotometer were highly correlated with the concentration of dietary AST [49].

## 5. Conclusions

This study suggests that dietary AST can effectively protect the skin from the effects of chronic UV exposure. It was confirmed that AST has antioxidation and anti-photoaging effects. Oral dietary AST is a promising anti-aging and antioxidant substance that can protect the skin from damage under UV. Our results demonstrated the potential of AST to be further developed as a pharmaceutical against photoaging.

## Figures and Tables

**Figure 1 biomedicines-08-00018-f001:** The effects of AST on the dorsal skin of ultraviolet (UV)-treated mice. (**A**) Photographs and replicas of the mouse dorsal skin. Scale bar = 50 μm. (**B**) Photographs of the mouse dorsal skin. Scale bar = 1 cm. (**C**) Histograms of the replica analysis. AST treatment improved the visible skin condition, reduced the proportion of wrinkle area to skin area (%), and the mean area of wrinkles in UV-induced mice. * *p* < 0.01 and # *p* < 0.01 vs. Control mice; # *p* < 0.01 vs. UV-treated mice.

**Figure 2 biomedicines-08-00018-f002:** The effect of AST on epidermal thickness in the dorsal skin of ultraviolet (UV)-induced mice. (**A**) Hematoxylin and eosin staining. Scale bar = 50 μm (**B**) Treatment with AST significantly suppressed the UV irradiation-induced increase in epidermal thickness. * *p* < 0.01 vs. control mice; # *p* < 0.01 vs. UV-induced mice.

**Figure 3 biomedicines-08-00018-f003:** The effect of AST on the density of collagen fibers (blue) in the dorsal skin of ultraviolet (UV)-induced mice. (**A**) Masson’s trichrome staining. Scale bar = 50 μm. (**B**) Treatment with AST significantly suppressed the UV irradiation-induced reduction in the collagen fiber density by preventing UV-induced loss of collagen fibers. * *p* < 0.05 vs. control mice; # *p* < 0.05 vs. UV-induced mice.

**Figure 4 biomedicines-08-00018-f004:** The effect of AST on changes in the density of capillary vessels in the dorsal skin of ultraviolet (UV)-induced mice. (**A**) AP staining. Arrow indicates the cross-section of a capillary. Scale bar = 200 μm. (**B**) Treatment with AST significantly suppressed the UV irradiation-induced regression in capillary vessels. * *p* < 0.01 vs. control mice; # *p* < 0.1 vs. UV-induced mice.

**Figure 5 biomedicines-08-00018-f005:** The effect of AST on ROS generation in the epidermis and capillary vessels in the dorsal skin of ultraviolet (UV)-induced mice. (**A**) Fluorescent staining. Scale bar = 200 μm. Arrows indicate an ROS expression region (the solid arrows point to epidermis, the dotted lines point to cross-sections of capillaries). (**B**) Treatment with AST significantly suppressed the UV irradiation-induced increase in ROS in the dorsal skin of UV-induced mice. * *p* < 0.01 vs. control mice; # *p* < 0.01 vs. UV-induced mice.

**Figure 6 biomedicines-08-00018-f006:** The correlation between the density of capillaries and density of wrinkles, density of collagen fiber, epidermal thickness, and ROS expression in the dorsal skin of ultraviolet (UV)-induced mice. (**A**) The density of capillaries is negatively correlated with the density of wrinkles. (**B**) The density of capillaries is positively correlated with the density of collagen fibers. (**C**) The density of capillaries is negatively correlated with the thickness of the epidermis. (**D**) The density of capillaries is negatively correlated with the reactive oxygen species (ROS) expression.

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
