# Peer review of "Protective Effects of Astaxanthin Supplementation against Ultraviolet-Induced Photoaging in Hairless Mice"

_biomedicines, 2020, doi:10.3390/biomedicines8020018_

Round 1

Reviewer 1 Report

This is an interesting and potentially valuable study of the effect of astaxanthin supplementation on the skin of UV-irradiated hairless mice as a model of human skin photoaging. Astaxanthin seemed to prevent irradiation-induced wrinkling, thickening of skin, loss of collagen fibers and of capillaries, possibly because it inhibited free radical formation. Unfortunately, the style of writing is so poor that sometimes the statements are opposite to authors’ intentions.

It looks like the authors never read what they wrote. There are fragments of unfinished sentences, repetitions, errors in spelling, punctuation, grammar and vocabulary, too numerous to point them all. Any word processing program should alert writers to many of these mistakes.

 Here are just a few examples of problems in this manuscript:

Abstract. Line 19-21. “…astaxanthin decreased… increase in collagen…attenuated….capillary number…” It is just the opposite.

Introduction. Line 3. There should be period after “hypodermis” and a new sentence starting from “Skin…” Page 2 line 2. What sort of sentence is “In addition.”? Line 11 “”accelerate health”?

 In general, the authors are too fond of “however”, ”therefore”, and similar words, and use them unnecessarily or in wrong places in whole article.

Materials and Methods. Page 2 line 28. “Astaxanthin was supplemented (by gavage? In oil?), not “implemented”. Were mice euthanized or just their skin was harvested under anesthesia (page 3 line 3, line 5)? Line 20 – "IMT i-solution" repeated twice. Line 25 – it should be “emit red light”.

Results. The legend of Fig. 1 is not under the figure but way down the page. While 1A has a 50 µm bar (not mentioned), 1B is supposed to have a scale bar = 200 µm (nonsense, 0.2 mm!) according to the legend. Where is the mean area of wrinkles mentioned in legend? “Percentage of wrinkle number” is meaningless, it should be “density of wrinkles (control = 100%)” i.e. number of wrinkles per unit area compared to control.

This “percentage” is used also in fig. 3B, 4B and 5B.

Page 5. Line 1-6. This paragraph is totally garbled, and the rest of description is not much better. Legend to Fig.2C “Histogram of… staining”?

Page 6. The legend of fig.3 is not under the figure but way down the page after a huge empty space.

Line 15. Why brackets (Figure A4)? Figure 4 legend is garbled; “The effect of astaxanthin on capillary vessels of the number changes…”? It is capillary density. The arrow is in 4A, not B, and points to a cross section of a capillary (not “arrow pointing”).

Figure 5 “epidermis and capillary vessels of ROS generation”?

Page 8. Line 10-11. It should be “density of wrinkles, density of capillaries and density of collagen fibers”, here and in Fig 6, in graphs and legend.

Discussion. Page 9, Line 11-13. Inattentive writing.

Carotenoids are present in human skin, but probably not in mice skin (poor absorption, conversion to vitamin A). The authors quote many references, but often do not specify species or tissue in question.

Page 10. Line 6. “We found that dietary astaxanthin accumulates in the skin…” The study did not find it, because the skin of hairless mice was not analyzed for astaxanthin content. In fact, such analysis would be very useful.

The Conclusions do not arise from the study but are sort of general statements.

The references are adequate but some titles are capitalized while other are not.

In summary, the manuscript requires a lot of work and attention from authors.

Author Response

Response to Reviewer 1

Dear :
Editor, biomedicines

We are submitting a revised version of our manuscript (biomedicines-676325) entitled “Protective Effects of Astaxanthin Supplementation against Ultraviolet-Induced Photoaging in Hairless Mice” by Xing Li et al. We have addressed all of the comments from the Editor and Referee on a point-by-point basis below. We appreciate the time that the Editor and Referee have taken to make helpful comments and we believe that the manuscript has been improved based on their comments.

We hope that we have satisfactorily addressed the Editor’s and Reviewer’s comments and that the manuscript is found to be acceptable for publication in biomedicines.

Sincerely yours,

Hidemi Fujino, Ph.D.

Professor, Department of Rehabilitation Science

Kobe University Graduate School of Health Sciences

Kobe, Japan

Comments and Suggestions for Authors

This is an interesting and potentially valuable study of the effect of astaxanthin supplementation on the skin of UV-irradiated hairless mice as a model of human skin photoaging. Astaxanthin seemed to prevent irradiation-induced wrinkling, thickening of skin, loss of collagen fibers and of capillaries, possibly because it inhibited free radical formation. Unfortunately, the style of writing is so poor that sometimes the statements are opposite to authors’ intentions.

It looks like the authors never read what they wrote. There are fragments of unfinished sentences, repetitions, errors in spelling, punctuation, grammar and vocabulary, too numerous to point them all. Any word processing program should alert writers to many of these mistakes.

Response: We thank the Reviewer for her/his positive comments. We hope that we have satisfactorily addressed the Reviewer’s comments.

Abstract::

Line 19-21. “…astaxanthin decreased… increase in collagen…attenuated….capillary number…” It is just the opposite.

Response: The Reviewer is correct. “The administration of AST reduced the UV-induced wrinkle formation and skin thickening and increased collagen fibers in the skin. AST supplementation also inhibited the generation of ROS, decreased wrinkle formation, reduced epidermal thickening,  and increased the density of capillaries in the skin..”

Introduction.

2 Line 3. There should be period after “hypodermis” and a new sentence starting from “Skin…

Response: We thank the Reviewer for the suggestion to edit the manuscript for grammar , we added the period after “hypodermis”.

Page 2 line 2. What sort of sentence is “In addition.”? Line 11 “”accelerate health”?

Response: We thank the Reviewer for the suggestion to edit the manuscript for grammar and English syntax. We hope that the revisions are satisfactory. we changed the sentences “In addition. The results are fine wrinkles due to diminished or defective synthesis of collagen and elastin in the dermis”to the “In time, UV irradiation results in fine wrinkles due to the diminished or defective synthesis of collagen and elastin in the dermis [4,8].”( New manuscript is page1, Line 38-39).

About Line 11 “”accelerate health”( New manuscript is page2, Line 3-4):

we changed the sentences“AST has been demonstrated to accelerate skin health through direct and downstream effects at different steps of the oxidative stress cascade in the skin [10]” to the “AST has been shown to improve dermal health by direct and downstream effects at different steps of the oxidative stress cascade, while simultaneously inhibiting inflammatory mediators [11]”.

In general, the authors are too fond of “however”, ”therefore”, and similar words, and use them unnecessarily or in wrong places in whole article.

Response: We thank the Reviewer for the suggestion to edit the manuscript for grammar and English syntax. We hope that the revisions are satisfactory. We have made an English language editing for the paper, to ensure the quality of English expressions.

Materials and Methods.

Page 2 line 28. “Astaxanthin was supplemented (by gavage? In oil?), not “implemented”.

Response: We agree with the Reviewer and thank her/him for pointing this out, This has now been stated in the Materials and Methods section as follows: “the mice were orally administered the olive oil only (dosages of olive oil 100 mg/kg body weight) by gavage daily. ”

Were mice euthanized or just their skin was harvested under anesthesia (page 3 line 3, line 5)?

Response: After the mice were anesthetized, their the skin was harvested. The mice were killed with an overdose of sodium pentobarbital sodium. This is now stated in the Materials and Methods 2.5 section as follows: “The mice then were killed with an overdose of sodium pentobarbital.”

Line 20 – "IMT i-solution" repeated twice.

Response: We thank the Reviewer for her/his suggestion. The "IMT i-solution" has been deleted.

Line 25 – it should be “emit red light”.

Response: Thanks for the reviewer's comments, Modifications were made as directed

This is now stated in the Materials and Methods 2.7 section as follows: “Dihydroethidium is a cell-permeable agent and interacts with nucleic acids to emit red light qualitatively detectable by fluorescence microscopy”.

Results:

The legend of Fig. 1 is not under the figure but way down the page.

 Response: We agree with the Reviewer and thank her/him for pointing this out, modifications were made as directed, we made a format adjustment

While 1A has a 50 µm bar (not mentioned), 1B is supposed to have a scale bar = 200 µm (nonsense, 0.2 mm!) according to the legend. Where is the mean area of wrinkles mentioned in legend?

Response: We agree with the Reviewer and thank her/him for pointing this out, we added a scale to the graph and we marked the square area of wrinkle in the figure as the mean area of wrinkle, and also explained it in the legend. This has now been stated in the legend section as follows:

Figure 1. The effects of AST on the dorsal skin of ultraviolet (UV)-treated mice. (A) Photographs and replicas of the mouse dorsal skin (magnified). Scale bar = 50 μm. (B) Photographs of the mouse dorsal skin, 1 cm above the line of the two iliac bones; 2 cm2 was taken from the spine as the centerline as the measurement area of wrinkles. Scale bar = 200 μm. (C) Histograms of the replica analysis. AST treatment improved the visible skin condition, reduced the percentage of wrinkles per unit area, and the mean area of wrinkles in UV-induced mice. * p < 0.01 and # p < 0.01 vs. Control mice; # p < 0.01 vs. UV-treated mice..”

“Percentage of wrinkle number” is meaningless, it should be “density of wrinkles (control = 100%)” i.e. number of wrinkles per unit area compared to control.

This “percentage” is used also in fig. 3B, 4B and 5B.

Response: We agree with the Reviewer and thank her/him for pointing this out. We changed the literal description of the Y-axis in the Figure 1C is “Density of wrinkles (control = 100%)”; in the Figure 3C is “ Density of collagen fibers (control=100%)”; in the Figure 4B is “ Density of capillaries (control=100%)”; in the Figure 5B is “ Density of ROS (control=100%)”.

We also changed the legends of Fig 1, Fig. 3, Fig 4 and Fig 5.

Page 5. Line 1-6. This paragraph is totally garbled, and the rest of description is not much

Response: We have revised the sentence in the “Page 5.Line 1-6” section as follows: “In this study, we investigated UV-induced skin photoaging using the HR-1 hairless mouse model. None of the treatments induced toxicity, which was determined by the loss of body weight (Figure 2C). To investigate the effects of AST on the photoaging of skin in vivo, the HR-1 mice were irradiated with UV. H&E staining revealed the effects of AST on histological changes in the dorsal skin (Figure 2A). As expected, the UV-induced mice had thicker epidermal layers than the mice that were not induced by UV. Nevertheless, the control mice and the AST-treated mice had thinner epidermal layers than the mice that were only irradiated with UV (Figure 2B).”

Legend to Fig.2C “Histogram of… staining”?

 Response: The reviewer makes a point. We added “μm” as the y-coordinate

Page 6. The legend of fig.3 is not under the figure but way down the page after a huge empty space.

Response: The Reviewer makes a good point, we made a format adjustment

Line 15. Why brackets (Figure A4)? Figure 4 legend is garbled; “The effect of astaxanthin on capillary vessels of the number changes…”? It is capillary density. The arrow is in 4A, not B, and points to a cross section of a capillary (not “arrow pointing”).

Response: Thanks to the comments of reviewer, We added arrows to the figure brackets in figureA, and explained them in the legend. We added the sentence in the legend section as follows: “Figure 4. The effect of AST on changes in the density of capillary vessels in the dorsal skin of ultraviolet (UV)-induced mice. (A) AP staining. Arrows indicate the cross section of a capillary. Scale bar = 200 μm. (B) Treatment with AST significantly suppressed the UV irradiation-induced regression in capillary vessels. * p < 0.01 vs. control mice; # p < 0.1 vs. UV-induced mice.”.

Figure 5 “epidermis and capillary vessels of ROS generation”?

ResponseWe added the arrows to the figure and explained them in the legend. We changed the sentence in the legend section as follows: “Figure 5. The effect of AST on ROS generation in the epidermis and capillary vessels in the dorsal skin of ultraviolet (UV)-induced mice. (A) Fluorescent staining. Scale bar = 200 μm. Arrows indicate a ROS expression region (where the solid arrow points is epidermis, where the dotted arrow points is the cross sections of capillary vessels). ”.

Page 8. Line 10-11. It should be “density of wrinkles, density of capillaries and density of collagen fibers”, here and in Fig 6, in graphs and legend.

Response: According to the Suggestions of reviewers, we have made modifications to all the articles and Fig6.

We changed the description in the legend section as follows: “Figure 6. The correlation between the density of capillaries and density of wrinkles, density of collagen-fiber, epidermal thickness, and ROS expression in the dorsal skin of ultraviolet (UV)-induced mice. (A) The density of capillaries is negatively correlated with the density of wrinkles. (B) The density of capillaries is positively correlated with the density of collagen-fibers. (C) The density of capillaries is negatively correlated with the thickness of the epidermis. (D) The density of capillaries is negatively correlated with the reactive oxygen species (ROS) expression.”

Page 9, Line 11-13. Inattentive writing.

Response: We thank the Reviewer for her/his positive comments. We have revised the sentence as follows: “(ii) AST supplementation decreased skin photoaging, which is characterized by reducing skin thickening and the density of wrinkling, and maintained ROS levels and the density of capillaries at control levels; (iii) changes in the density of capillaries were observed in the skin of hairless mice, which has a close correlation with the density of wrinkles, density of collagen, epidermal thickness, and ROS levels. Thus, AST treatment was demonstrated as a protective therapy for skin photoaging”

Carotenoids are present in human skin, but probably not in mice skin (poor absorption, conversion to vitamin A).

Response: Petri and Lundebye reported that the accumulation of dietary astaxanthin in hairless skin of the tail was much higher than those in other tissues. And color changes on the tail skin by the reflectance measurements using a portable spectrophotometer was highly correlated with the concentration of dietary astaxanthin(Petri D, Lundebye AK. Tissue distribution of astaxanthin in rats following exposure to graded levels in the feed. Comp Biochem Physiol C. 2007; 145: 202–209.)

The authors quote many references, but often do not specify species or tissue in question.

Response: The Reviewer makes a good point. For cited references, we describe especify tissues or species mentioned in the literatur. For example, “UV radiation has been reported to promote the activation of enzymes that destroy elastic fibers and collagen, making the skin more prone to wrinkling in human facial skin after UV irradiation [34]”;  “UV radiation has been reported to promote the activation of enzymes that destroy elastic fibers and collagen, making the skin more prone to wrinkling in human facial skin ”; “ In addition, AST increased collagen content through inhibition of MMP-1 and MMP-3 protein expression in human dermal fibroblasts [36] ”; “In the human brain microvascular endothelial cell line and rat aortic smooth muscle cell, it also has been reported that capillary proliferation is related to ROS level regulation, in angiogenesis, reactive oxygen species ROS signals regulate the formation of new blood vessels [49]” and so on.

Page 10. Line 6. “We found that dietary astaxanthin accumulates in the skin…” The study did not find it, because the skin of hairless mice was not analyzed for astaxanthin content. In fact, such analysis would be very useful

Response: We found that dietary AST may prevent the effects of UV irradiation on photoaging, as well as on the dermal capillaries. However, the cumulative intake of AST in the skin and blood was not measured, nor did we ascertain the mechanism by which AST inhibits capillary regression. Petri and Lundebye have reported that the accumulation of dietary AST in hairless skin of the tail was much higher than those in other tissues. Color changes on the tail skin by the reflectance measurements using a portable spectrophotometer was highly correlated with the concentration of dietary AST[50].”

The Conclusions do not arise from the study but are sort of general statements.

Response: We thank the Reviewer for her/his positive comments, We made statements like this about the conclusion:“This study has proved that dietary AST can effectively protect the skin from the effects of chronic UV exposure. It was confirmed that AST has anti-oxidation and anti-photoaging effects. Oral dietary AST is a promising anti-aging and antioxidant substance that can protect the skin from damage under UV. Our results demonstrated the potential of AST to be further developed as a pharmaceutical against photoaging.”

The references are adequate but some titles are capitalized while other are not.

Response: The Reviewer makes a good point. We made the right adjustments to the references.

Reviewer 2 Report

The manuscript entitled “Protective Effects of Astaxanthin Supplementation against Ultraviolet-Induced Photoaging in Hairless Mice” by Li et al. is an interesting and comprehensive article.  Nevertheless, the Authors did not escape some ambiguities in the text. After careful analysis of the manuscript, I have a few general comments and  suggestions, presented below:

The list of all Abbreviation should be added. The Introduction should explain why Author choose the 100 mg/kg/daily administration astaxanthin diet. Page 2 line 6-7 the sentence "Astaxanthins are characterized by a great variety of advantageous biological activities, promoting favorable outcomes" suggests to improve. Throughout the body text, I found some language and stylish errors. All ambiguities should be corrected. Materials and methods - should be improved.

I suggest adding some more information about animals - why males were used?;  what about humidity or bedding and enrichment and commercial diet? - I suggest add codes and supplier companies.

Were the animals' food and water consumption recorded, if so during what period?

I suggest subsection: Animal Experiments or Animals and Diets or Astaxanthin Source, Animals, and Experimental Design, and adding all information about the animal experiment – for example in: Nutrients 2019, 11(6), 1244; https://doi.org/10.3390/nu11061244.

Page 3 line 13-14: The information needs clarification

"..... under anesthesia at the end of the experiment at an age of 17 weeks? Was the animal still bred and exposed to UV?

Page 3 line 27-28 and 34 and 39: this paper not in

the incorrect citation; these papers (references) are not in the Reference.

Why the Scheffe test was used in the statistical analysis - please explain and add what program was used

Results,

The figure should be combined with its title - please change - the order first description of results then figure 1.

The figure shows Scale bar = 50 μm in the description Scale bar = 200 μm. Should this be corrected / clarified?

 The description of subchapter 3.2 requires clarification and improve.

Can toxicity, including astaxanthin, be determined only by the loss of body weight?

Section 3.6 also needs to be improved - title and description of Figure 6.

Discussion,

The discussion should be more focused on the results obtained. Consider sorting out the discussion and dividing it into subsections.

According to my opinion, the authors should extend discussion concerning Strengths and Limitations to this study.

Conclusions,

The conclusions presented do not apply to this study.

The conclusions should respond to the research goal and research hypotheses.

This should be explained.

The reviewer has made some mistakes about references cited in the manuscript. Please unification the reference.

The authors should consider the content of the comprehensive review published by Davinelli et al (2018) in Nutrients on the protective effects of astaxanthin in skin health (Nutrients 2018, 10, 522; doi:10.3390/nu10040522).

References should be improved in line with the publisher's requirements  - e.g. 13.

It is also worth considering the following paper:

Meephansan, J.; Rungjang, A.; Yingmema, W.; Deenonpoe, R.; Ponnikorn, S. Effect of astaxanthin on cutaneous wound healing. Clin. Cosmet. Investig. Dermatol. 2017, 10, 259–265. Hart, P.H.; Norval, M. Ultraviolet radiation-induced immunosuppression and its relevance for skin carcinogenesis. Photochem. Photobiol. Sci. 2017. Poljšak, B.; Dahmane, R.G.; Godi´c, A. Intrinsic skin aging: The role of oxidative stress. Acta Dermatovenerol. Alp Pannonica Adriat 2012, 21, 33–36. Suganuma, K.; Nakajima, H.; Ohtsuki, M.; Imokawa, G. Astaxanthin attenuates the UVA-induced up-regulation of matrix-metalloproteinase-1 and skin fibroblast elastase in human dermal fibroblasts. J. Dermatol. Sci. 2010, 58, 136–142.

Author Response

Response to Reviewer 2

Dear :
Editor, biomedicines

We are submitting a revised version of our manuscript (biomedicines-676325) entitled “Protective Effects of Astaxanthin Supplementation against Ultraviolet-Induced Photoaging in Hairless Mice” by Xing Li et al. We have addressed all of the comments from the Editor and Referee on a point-by-point basis below. We appreciate the time that the Editor and Referee have taken to make helpful comments and we believe that the manuscript has been improved based on their comments.    

We hope that we have satisfactorily addressed the Editor’s and Reviewer’s comments and that the manuscript is found to be acceptable for publication in biomedicines.

Sincerely yours,

Hidemi Fujino, Ph.D.

Professor, Department of Rehabilitation Science

Kobe University Graduate School of Health Sciences

Kobe, Japan

The manuscript entitled “Protective Effects of Astaxanthin Supplementation against Ultraviolet-Induced Photoaging in Hairless Mice” by Li et al. is an interesting and comprehensive article.  Nevertheless, the Authors did not escape some ambiguities in the text. After careful analysis of the manuscript, I have a few general comments and suggestions, presented below:

Response: We thank the Reviewer for her/his positive comments. We hope that we have satisfactorily addressed the Reviewer’s comments.

1.The list of all Abbreviation should be added.

Response: As the Reviewer has suggested, it has been added the list of all Abbreviation at the end of the paper as follows:

“The list of all abbreviations:

Astaxantin-------------------------------------------------------AST

Ultraviolet--------------------------------------------------------UV

Generation of reactive oxygen species -------------------ROS

Alkaline phosphatase ----------------------------------------AP

Minimal erythema dose--------------------------------------MED

Hematoxylin staining-----------------------------------------H&E

Fluorescent dihydroethidium ---------------------------------DHE”

The Introduction should explain why Author choose the 100 mg/kg/daily

Response: We thank the Reviewer for the suggestion. Astaxanthin supplementation was performed by the method previously described. Kanazashi et al. (Acta Physiol, 2013, Exp Physiol, 2014) reported that astaxanthin was administered orally by a feeding tube twice a day (50 mg/kg, with a 6 h interval between the two doses; total of 100 mg/kg/day). Astaxanthin supplementation resulted in preventing capillary regression in atrophied muscle. Maezawa et al. (J physiol Sci, 2017) also reported that astaxanthin supplementation was performed by orally at 100 mg/kg/day by a feeding tube in joint immobilization model of rats. Astaxanthin has been shown to reduce ROS production in immobilized muscle. Therefore, we were performed by same method and dose of astaxanthin.

Page 2 line 6-7 the sentence "Astaxanthins are characterized by a great variety of advantageous biological activities, promoting favorable outcomes" suggests to improve.

Response: We thank the Reviewer for her/his positive comments. we deleted the sentence "Astaxanthins are characterized by a great variety of advantageous biological activities, promoting favorable outcomes". then added the accurate description to the introduction in the page1,Line 42 of the new manuscripts as follows: “AST possesses many highly potent pharmacological effects”.

Throughout the body text, I found some language and stylish errors. All ambiguities should be corrected. Materials and methods - should be improved.

Response: We thank the Reviewer for the suggestion to edit the manuscript for language and stylish errors. We hope that the revisions are satisfactory. We have made an English language editing for the paper, to ensure the quality of English expressions.

We have made a language editing in Materials and methods

I suggest adding some more information about animals - why males were used?; what about humidity or bedding and enrichment and commercial diet? - I suggest add codes and supplier companies.

Response: The reviewer makes a good point. The exclusion of female rodents from biomedical research is well documented and persists in large part due to perceptions that ovulatory cycles render female traits more variable than reliable data. These beliefs are not empirically based. The cutaneous microvascular response and skin condition is involved the phases of the menstrual cycle (Benjiamin Smarr et al. Male and female mice show equal variability in food intake across 4-day spans that encompass estrous cycles. Plos one 2019, doi.org/10.1371/journal.pone.0218935). Thus, to ensure the accuracy of the experiment, we used male mice as samples.
    The mice were housed individually under a 12 h/12 h light/dark cycle at room temperature (22 ± 2°C) and 50% relative humidity. Wood chips were used as bedding in the cages and replaced every 3 days. Hairless mice (HR-1) were obtained fromJapan SLC, Inc.; they were fed a commercial diet (CE-2, CLEA Japan, Inc.) and allowed access to tap water ad libitum throughout the study. AST oil was provide by Fuji Chemical Industry Co. Ltd, Toyama, Japan. These are added to the Materials and Mehods.
The codes and supplier company have been also added to the Materials and Mehods

Were the animals' food and water consumption recorded, if so during what period?

Response: The Reviewer makes a good point. The consumption recorded of water and food was recorded once a week from the beginning of the experiment to the end

I suggest subsection: Animal Experiments or Animals and Diets or Astaxanthin Source, Animals, and Experimental Design, and adding all information about the animal experiment – for example in: Nutrients 2019, 11(6), 1244; https://doi.org/10.3390/nu11061244.

Response: We agree with the Reviewer and thank her/him for pointing this out . Animals and Diets or Astaxanthin Source and Experimental Design, and adding all information about the animal experiment have been added to the paper, then indicated the name of the manufacturer. we added the sentences as follows:

“2.1. Animals and Astaxanthin Source

The mice were housed individually under a 12 h/12 h light/dark cycle at room temperature (22 ± 2°C) and 50% relative humidity. Wood chips were used as bedding in the cages and replaced every 3 days. Hairless mice (HR-1) were obtained fromJapan SLC, Inc.; they were fed a commercial diet (CE-2, CLEA Japan, Inc.) and allowed access to tap water ad libitum throughout the study. AST oil was provide by Fuji Chemical Industry Co. Ltd, Toyama, Japan.

2.2. Astaxanthin Treatment and Experimental Design

The HR-1 hairless mice were randomly divided into three groups (n = 8 per group): (1) control, olive oil alone (CON), (2) UV-induced plus olive oil (UV), (3) UV-induced plus AST (AstaReal Oil 50F, Fuji Chemical Industry Co. Ltd, Toyama, Japan). In the CON group, the mice were orally administered the olive oil only (dosages of olive oil 100 mg/kg body weight) by gavage daily. In the UV and AST group, the mice were orally supplemented with the mixture of AST (dosages of AST 100 mg/kg body weight). This study was approved by the Institution Animal Care and Use Committee; the experimental protocol followed the Kobe University Animal Experimentation Regulations (Kobe, Japan). All experiment and animal care programs were managed in conformance with the Guide for the Care and Use of Laboratory Animals published by the US National Institutes of Health (NIH publication no. 85-23, revised 996).”

Page 3 line 13-14: The information needs clarification

"..... under anesthesia at the end of the experiment at an age of 17 weeks? Was the animal still bred and exposed to UV?

Response: We agree with the Reviewer and thank her/him for pointing this out.

The mice were anesthetized 16 week and tissue was removed, then were killed, we added the sentence as follow: “The dorsal skin of the mice located between the ilia was harvested at the end of the experiment, at 16 weeks after the application of anesthesia; the mice were then killed with an overdose of sodium pentobarbital”.

Page 3 line 27-28 and 34 and 39: this paper not in the incorrect citation; these papers (references) are not in the Reference.

Response: The reviewer makes a good point. The reference has been modified

Why the Scheffe test was used in the statistical analysis - please explain and add what program was used.

Response: We thank the Reviewer for her/his positive comments. In our study, there are three groups of average data, which need to be compared between groups. When the differences of more than three average values need to be compared, The T-estssimple is used is not reliable, so we used the Scheffe test in multiple Comparison procedures.

This has now been stated in the 2.8 section as follows: “The data are presented as the mean ± SD. The Statistical analyses were carried out using one-way ANOVA, followed by the Scheffe’s post-hoc test. The SigmaStat statistical program (version11.2; Systat Software, San Jose, CA, USA) was used for statistical analysis of the data, Differences are considered significant at p < 0.05.

Results:

The figure should be combined with its title - please change - the order first description of results then figure 1.

Response: Thanks for the Suggestion of reviewes there have been modified

The figure shows Scale bar = 50 μm in the description Scale bar = 200 μm. Should this be corrected / clarified?

Response: The reviewer is correct, we added the scale bar = 200 μm in the figure 1B.

We corrected the legdend of figure 1 as follows:

Figure 1. The effects of AST on the dorsal skin of ultraviolet (UV)-treated mice. (A) Photographs and replicas of the mouse dorsal skin (magnified). Scale bar = 50 μm. (B) Photographs of the mouse dorsal skin, 1 cm above the line of the two iliac bones; 2 cm2 was taken from the spine as the centerline as the measurement area of wrinkles. Scale bar = 200 μm. (C) Histograms of the replica analysis. AST treatment improved the visible skin condition, reduced the percentage of wrinkles per unit area, and the mean area of wrinkles in UV-induced mice. * p < 0.01 and # p < 0.01 vs. Control mice; # p < 0.01 vs. UV-treated mice.”

The description of subchapter 3.2 requires clarification and improve.

Response: We thank the Reviewer for the suggestion. We corrected the description of 3.2 as follows: “In this study, we investigated UV-induced skin photoaging using the HR-1 hairless mouse model. None of the treatments induced toxicity, which was determined by the loss of body weight (Figure 2C). To investigate the effects of AST on the photoaging of skin in vivo, the HR-1 mice were irradiated with UV. H&E staining revealed the effects of AST on histological changes in the dorsal skin (Figure 2A). As expected, the UV-induced mice had thicker epidermal layers than the mice that were not induced by UV. Nevertheless, the control mice and the AST-treated mice had thinner epidermal layers than the mice that were only irradiated with UV (Figure 2B)”.

Can toxicity, including astaxanthin, be determined only by the loss of body weight?

Response: Toxicity evaluation in addition to the weight, and, of course, the liver and kidney Toxicity, biochemical examination, etc. But, the safety of astaxanthin has been reported (Ref: 1.Stewart JS, Lignell Å, Pettersson A, et al. Safety assessment of astaxanthin-rich microalgae biomass: Acute and subchronic toxicity studies in rats [J]. Food Chem Toxicol, 2008, 46(9): 3030-3036.) 2. Katsumata T, Ishibashi T, Kyle D. A sub-chronic toxicity evaluation of a natural astaxanthin-rich carotenoid extract of Paracoccus carotinifaciens in rats [J]. Toxicol Rep, 2014, (1): 582-588.

Section6 also needs to be improved - title and description of Figure 6.

Response: We thank the Reviewer for her/his positive comments. This has now been stated in the 3.6 section as follows: “Correlation between the density of Capillaries and Epidermal Thickness, Density of Collagen , and ROS Expression in the Dorsal Skin of UV-Induced Mice” as a title.

Description of the Section 3.6 is as follows: “We also investigated the relationship between capillaries, wrinkles, epidermal thickness, collagen, and ROS. We found that the density of capillaries is negatively correlated with the density of wrinkles, the thickness of epidermis and the ROS levels (Figure 6A, C, D). The density of capillaries is positively correlated with the density the collagen fibers (Figure 6B). Evidently, changes in the density of capillaries have an important effect on the photoaging of dorsal skin in UV-induced mice.”

The description of the legend for the Figure 6 is as follow:

Figure 6. The correlation between the density of capillaries and density of wrinkles, density of collagen-fiber, epidermal thickness, and ROS expression in the dorsal skin of ultraviolet (UV)-induced mice. (A) The density of capillaries is negatively correlated with the density of wrinkles. (B) The density of capillaries is positively correlated with the density of collagen-fibers. (C) The density of capillaries is negatively correlated with the thickness of the epidermis. (D) The density of capillaries is negatively correlated with the reactive oxygen species (ROS) expression.”

Discussion:

The discussion should be more focused on the results obtained. Consider sorting out the discussion and dividing it into subsections.

Response: We thank the Reviewer for her/his positive comments: We adjusted the< discussion section> and discussed the study results respectively.

According to my opinion, the authors should extend discussion concerning Strengths and Limitations to this study.

Response: We agree with the Reviewer and thank her/him for pointing this out. concerning Strengths and Limitations to this study, It has been explained in the paper

For example, concerning Strengths: “. We found that the density of capillaries is negatively correlated with the density of wrinkles, the thickness of epidermis and ROS levels. We also showed that the density of capillaries is positively correlated with the density the collagen fibers.” in the page8, Line 12-14 in the new manuscript.

Limitations: “...the relationship between the mechanism of capillary density degradation and AST was not explained in this experiment and would therefore benefit from further study”.

Conclusions:

The conclusions presented do not apply to this study.The conclusions should respond to the research goal and research hypotheses. This should be explained.

ResponseWe thank the Reviewer for her/his positive comments. This has now been stated in the conclusion section as follows: “the relationship between the mechanism of capillary density degradation and AST was not explained in this experiment and would therefore benefit from further study.”

The reviewer has made some mistakes about references cited in the manuscript. Please unification the reference.

Response: The reviewer makes a good point. We have modified the reference format

The authors should consider the content of the comprehensive review published by Davinelli et al (2018) in Nutrients on the protective effects of astaxanthin in skin health (Nutrients 2018, 10, 522; doi:10.3390/nu10040522).

Response Thank you very much for the suggestions of reviewers. We have modified the reference format and the article has been cited in this paper, the number is [23] in the reference.

It is also worth considering the following paper:

Meephansan, J.; Rungjang, A.; Yingmema, W.; Deenonpoe, R.; Ponnikorn, S. Effect of astaxanthin on cutaneous wound healing. Clin. Cosmet. Investig. Dermatol. 2017, 10, 259–265.

Response Thanks for reviewer’s suggestions,this article has been cited in this paper, the number is [33] in the reference.

Hart, P.H.; Norval, M. Ultraviolet radiation-induced immunosuppression and its relevance for skin carcinogenesis. Photochem. Photobiol. Sci. 2017.

Response The article has been cited in this pape, the number is [6] in the referrence.

Poljšak, B.; Dahmane, R.G.; Godi´c, A. Intrinsic skin aging: The role of oxidative stress. Acta Dermatovenerol. Alp Pannonica Adriat 2012, 21, 33–36.  

Response: The article has been cited in this paper,the number is [24] in the referrence.

Suganuma, K.; Nakajima, H.; Ohtsuki, M.; Imokawa, G. Astaxanthin attenuates the UVA-induced up-regulation of matrix-metalloproteinase-1 and skin fibroblast elastase in human dermal fibroblasts. J. Dermatol. Sci. 2010, 58, 136–142.

Response The article has been cited in this paper, the number is [34] in the referrence.

Round 2

Reviewer 1 Report

The style of the manuscript is greatly improved, as well as the discussion and conclusions. However, it still requires considerable work. The authors were not very careful in their revisions.

Below is the list of necessary corrections. The authors should not assume that the list is complete, the reviewer may have overlooked some mistakes.

Page 1 line 35. “…ofenes”? Is it “of genes”? Line 36 – space missing before “It…” Line 37 – remove comma after “stress”, insert comma before “as well as”, it should be “are induced”, not “is”. Line 40 – insert comma after “salmon”.

Page 2 line 3. Insert comma after “activity”. Line 29 – dosage (sing.)

Page 3 line 28. It should be “with ImageJ using…” not “using” twice. Line 35. This study used skin, not soleus muscle! Line 40 –it should be “statistical”.

Page 4 line 8. It should be “thick and deep.” In Figure 1, a C is missing next to the histogram. It should be AST+UV in the histogram and in photographs A and B. The legend is muddled – photograph B shows nearly the whole back of a mouse, what is “1 cm above the line of iliac bones”? A square of 2 cm2 centered on the spine (B, Ast+UV photo) was apparently used to estimate the area of wrinkles. The term “area of wrinkles” seems to be interchangeable with “density of wrinkles” – it should be explained in Methods (not “percentage of wrinkles per unit area” but “proportion of wrinkle area to skin area (%)”. Line 22-23 and next page 23. It should not be “expected”, that is a bias of researchers! This sentence and the next are saying essentially the same thing. Unnecessary “nevertheless”. However the results are not quantitatively described here and in all Results (statistical significance, magnitude of differences).

Page 5. Fig. 2. See above for AST+UV in the photos and graphs. The graph C should not be in this figure, nor the evaluation of the lack of treatment toxicity in this subchapter 3.2. It should be in a separate figure 1 preceding other results. It could be inserted in Methods (2.3), as “Body weight of HR-1 hairless mice undergoing different treatments, etc." The remaining legend of Fig. 2 is still not correct. B - is a histogram of epidermal thickness (not of staining!). Again, no quantitative evaluation in the text. Line 13 – again, too many “nevertheless”.

Page 6. Fig. 3. See above for AST+UV in the photo and graph. Insert “(blue)” after “fibers” in line 2.  Line 7 - it should be “changes I the density of capillaries”. Lines 9-10 – terribly repetitive sentence (“we found that results show...” is totally unnecessary. However, quantification of the results in text is rather desirable, but it is missing, although the capillary density doubled with AST+UV treatment compared to control, while it dropped by half with irradiation (fig 4).

Page 7. Fig. 4. See above for AST+UV in the photos and graph. On B graph Y axis it should be “Capillary density (control = 100%)” – not corrected despite previous reviewer’s comments. Line 3 – it should be “Arrow indicates…”. Line 8 – a space is missing after HR-1.

Page 8. Fig. 5. See above for AST+UV in the photos and graph. On B graph Y axis it should be “ROS expression (% control)”. Line 4 – it should be (the solid arrow points to epidermis, the dotted line points to cross-sections of capillaries).

Page 8. Line 17 – comma after “organs”.

Page 9. Line 15. It should be AST+UV. Line 43. It should be “soleus muscle of rats…” Line 44 – remove “in atrophied soleus muscle of rats.”

Page 10. The lines 10-14 should be moved to Conclusions, before line 24. Line 24. It should be “This study suggest (or indicates)…””Has proved” is too boastful. Line 17. The article by Petri and Lundebye is missing from the References. Line 45. It is References (pl.).

As shown above, the manuscript still requires numerous corrections before it may be published.

Author Response

Response to reviewer #1

Page 1 line 35. “…ofenes”? Is it “of genes”? Line 36 – space missing before “It…” Line 37 – remove comma after “stress”, insert comma before “as well as”, it should be “are induced”, not “is”. Line 40 – insert comma after “salmon”.

Response:  We agree with the Reviewer and thank her/him for pointing this out. We revised the manuscript.

Page 2 line 3. Insert comma after “activity”. Line 29 – dosage (sing.)

Response:  We agree with the Reviewer and thank her/him for pointing this out. We revised the manuscript.

Page 3 line 28. It should be “with ImageJ using…” not “using” twice. Line 35. This study used skin, not soleus muscle! Line 40 –it should be “statistical”.

Response:  We agree with the Reviewer and thank her/him for pointing this out. We revised the manuscript.

Page 4 line 8. It should be “thick and deep.” In Figure 1, a C is missing next to the histogram. It should be AST+UV in the histogram and in photographs A and B.

Response: Thanks for the reviewer's comments. We added the “AST+UV” in the histogram and in photographs A and B.

The legend is muddled – photograph B shows nearly the whole back of a mouse, what is “1 cm above the line of iliac bones”? A square of 2 cm2 centered on the spine (B, Ast+UV photo) was apparently used to estimate the area of wrinkles. The term “area of wrinkles” seems to be interchangeable with “density of wrinkles” – it should be explained in Methods (not “percentage of wrinkles per unit area” but “proportion of wrinkle area to skin area (%)”.

 Response: We agree with the Reviewer and thank her/him for pointing this out. We added the description to the 2.4 method:“...To estimate the proportion of wrinkle area to skin area , a square of 2 cmcentered on the spine was taken”. The new photographs were replaced in Fig 1B and the legend was described as follows: “Figure 1. The effects of AST on the dorsal skin of ultraviolet (UV)-treated mice. (A) Photographs and replicas of the mouse dorsal skin. Scale bar = 50 μm. (B) Photographs of the mouse dorsal skin. Scale bar = 1 cm. (C) Histograms of the replica analysis. AST treatment improved the visible skin condition, reduced the proportion of wrinkle area to skin area (%) ...”.

Line 22-23 and next page 2-3. It should not be “expected”, that is a bias of researchers! This sentence and the next are saying essentially the same thing. Unnecessary “nevertheless”. However, the results are not quantitatively described here and in all Results (statistical significance, magnitude of differences).

Response: Thank you for the reviewer’s comments. We deleted the “expercted” and changed the Line 22-23 as follows; “while those of the UV-induced mice were thick and deep. Moreover, the UV-induced mice showed an increase in the percentage (approximately 1.6 times) area of wrinkles compared with the control mice (Figure 1C). These changes were reduced by AST treatment, which was been reduced by 50% compared with that of the UV group. Analysis showed that AST reduces UV-induced wrinkle formation”.

Page 5. Fig. 2. See above for AST+UV in the photos and graphs. The graph C should not be in this figure, nor the evaluation of the lack of treatment toxicity in this subchapter 3.2. It should be in a separate figure 1 preceding other results. It could be inserted in Methods (2.3), as “Body weight of HR-1 hairless mice undergoing different treatments," The remaining legend of Fig. 2 is still not correct. B - is a histogram of epidermal thickness (not of staining!). Again, no quantitative evaluation in the text. Line 13 – again, too many “nevertheless”.

 Response: We agree with the Reviewer and thank her/him for pointing this out. We deleted the Fig2c, It is inserted in Methods (2.3), added the sentence as follows: ”... Body weight of HR-1 hairless mice undergoing different treatments, the food and water consumption were recorded every day”. We changed the legend of Fig 2 as follow: “Figure 2. The effect of AST on epidermal thickness in the dorsal skin of ultraviolet (UV)-induced mice. (A) Hematoxylin and eosin staining. Scale bar = 50 μm (B) Treatment with AST significantly suppressed the UV irradiation-induced increase in epidermal thickness. * p < 0.01 vs. control mice; # p < 0.01 vs. UV-induced mice”. In addition, we stated about body weight in Results as follows; “ In the present study, body weights were measured for determining the toxicity of the treatment, we found that there was no significant difference in body weight between CON group, UV group and UV +AST group.”

About the “nevertheless”, we revised to other words. We hope that the revisions are satisfactory. We have made an English language editing for the paper, to ensure the quality of English expressions.

Page 6. Fig. 3. See above for AST+UV in the photo and graph. Insert “(blue)” after “fibers” in line 2.  Line 7 - it should be “changes I the density of capillaries”. Lines 9-10 – terribly repetitive sentence (“we found that results show...” is totally unnecessary. However, quantification of the results in text is rather desirable, but it is missing, although the capillary density doubled with AST+UV treatment compared to control, while it dropped by half with irradiation (Fig 4).

Response: We agree with the Reviewer and thank her/him for pointing this out. We revised the manuscript as follows: “In order to investigate the changes of the density capillaries in the dorsal skin of UV-induced mice, AP staining was used to evaluate the presence and distribution of capillaries. As shown in Figure 4A, we also found that the results show that the density of capillary vessels is reduced in the dorsal skin of UV-induced mice. The density of capillary vessels of UV group decreased by approximately 52% compared with that of the CON group. Whereas, the changes can be reversed by oral AST, the density of capillary vessels in UV+AST group increased by approximately 4 times compared with that in the UV group. This result shows that AST attenuates UV-induced capillary regression.”

Page 7. Fig. 4. See above for AST+UV in the photos and graph. On B graph Y axis it should be “Capillary density (control = 100%)” – not corrected despite previous reviewer’s comments. Line 3 – it should be “Arrow indicates…”. Line 8 – a space is missing after HR-1.

 Response: We agree with the Reviewer and thank her/him for pointing this out.

Page 8. Fig. 5. See above for AST+UV in the photos and graph. On B graph Y axis it should be “ROS expression (% control)”. Line 4 – it should be (the solid arrow points to epidermis, the dotted line points to cross-sections of capillaries). Page 8. Line 17 – comma after “organs”.

 Response: Response: We thank the Reviewer for the suggestion to edit the manuscript for grammar and English syntax. We revised the manuscript.

Page 9. Line 15. It should be AST+UV. Line 43. It should be “soleus muscle of rats…” Line 44 – remove “in atrophied soleus muscle of rats.”

 Response: We agree with the Reviewer and thank her/him for pointing this out. We revised the manuscript.

Page 10. The lines 10-14 should be moved to Conclusions, before line 24. Line 24. It should be “This study suggest (or indicates)…””Has proved” is too boastful. Line 17. The article by Petri and Lundebye is missing from the References. Line 45. It is References (pl.)

Response: We thank the Reviewer for the suggestion to edit the manuscript for grammar and English syntax. We revised the manuscript and added the reference in paper (the number is 49).